# Evaluation of Static DNA Ploidy Analysis Using Conventional Brush Biopsy-Based Cytology Samples as an Adjuvant Diagnostic Tool for the Detection of a Malignant Transformation in Potentially Oral Malignant Diseases: A Prospective Study

**DOI:** 10.3390/cancers14235828

**Published:** 2022-11-26

**Authors:** Natalie Bechstedt, Natalia Pomjanski, Martin Schramm, Torsten W. Remmerbach

**Affiliations:** 1Department of Oral and Maxillofacial and Facial Plastic Surgery, Section of Clinical and Experimental Oral Medicine, University Hospital Leipzig, Liebigstraße 10–14, 04103 Leipzig, Germany; 2Department of Cytopathology, Institute of Pathology, Heinrich Heine University, 40225 Düsseldorf, Germany

**Keywords:** aneuploidy, oral cancer, DNA image cytometry, oral potentially malignant disorders, brush biopsy, mass screening tool

## Abstract

**Simple Summary:**

Oral cancer is one of the most common malignant diseases worldwide. What is particularly challenging is not the visual detection of a change in the oral cavity, since this is a clearly visible area, but rather the classification between benign and malignant lesions. With brush biopsies and the evaluation of the DNA content of cells using image cytometry, an examination method was found that is non-invasive and offers a patient-friendly possibility to clarify changes in the oral mucosa. The application of the method is controversial for various reasons. With a cohort of 602 cases, we have examined the effectiveness of this examination method on different oral potentially malignant disorders as well as on a large number of confirmed squamous cell carcinomas. In our study, we confirm the efficiency of DNA ploidy analysis to improve the diagnostic accuracy of conventional cytology.

**Abstract:**

Background: The accuracy of DNA image cytometry as an investigation method for potentially malignant disorders of the oral cavity is currently still a subject of controversy, due to inconsistently applied definitions of DNA aneuploidy, small cohorts and different application techniques of the method. The aim of this study was to examine the accuracy of the method as a supplementary diagnostic tool in addition to the cytological examination using internationally consented definitions for DNA aneuploidy. Methods: A total of 602 samples from 467 patients with various oral lesions were included in this prospective study. Brush biopsies from each patient were first cytologically examined and categorized by a pathologist, second evaluated using DNA image cytometry, and finally compared to either histological biopsy result or clinical outcome. Results: Using the standard definition of DNA aneuploidy, we achieved a sensitivity of 93.5%, a positive predictive value for the detection of malignant cells of 98.0%, and an area under the curve of 0.96 of DNA ploidy analysis for the detection of severe oral epithelial dysplasia, carcinoma in situ or oral squamous cell carcinoma. Importantly, using logistic regression and a two-step model, we were able to describe the increased association between DNA-ICM and the detection of malignant cells (OR = 201.6) as a secondary predictor in addition to cytology (OR = 11.90). Conclusion: In summary, this study has shown that DNA ploidy analysis based on conventional specimens of oral brush biopsies is a highly sensitive, non-invasive, patient-friendly method that should be considered as an additional diagnostic tool for detecting malignant changes in the oral cavity.

## 1. Introduction

Cancer of the lip and oral cavity is one of the most common malignancies worldwide. In 2020, the World Health Organization recorded 377,713 new cases and 177,757 new deaths for lip and oral cavity cancer [1]. In total, 65.8% of these cases are recorded in Asia, with the highest estimated incidence rates in Papua New Guinea, India, Pakistan and Bangladesh [1].

Histologically, the most common is oral squamous cell carcinoma (OSCC), and main risk factors include tobacco use, chewing betel nut, and alcohol consumption [2].

Oral potentially malignant disorders (OPMD), including mainly leukoplakia, erythroplakia and oral lichen planus, are mucosal diseases that have a risk of malignancy being present at the time of initial diagnosis or a future date [3]. The majority of these OPMDs may not progress to OSCC, yet they provide a field of abnormalities in which possible cancer development is more likely than in patients without such disorders [4]. Overall, oral lesions can be easily detected visually during dental screening, however, classification between benign lesion, OPMD and OSCC is challenging.

At an early stage of diagnosis, the 5-year survival rate of OSCC is 85%; however, only 28% of oral cancers are diagnosed at this point. Almost 50% of cases are not recognized until the tumor is in an advanced stage and has infiltrated the locoregional lymph nodes. In this case, the 5-year survival rate drops to 68%. Furthermore, 18% of the tumors are not recognized until metastases spread throughout distant parts, causing the 5-year survival rate to drop to 40% [5].

In various cytogenetic carcinogenesis theories of preneoplastic and neoplastic cells, the transition from stable diploid to unstable aneuploid cells is discussed as one of the main causes. The progression of malignant cells arises from cancer-specific primary, secondary, and tertiary chromosomal changes. Secondary aberrations can be detected by collecting cells, through non-invasive brush biopsies, whose DNA content is measured by DNA image cytometry (DNA-ICM). DNA aneuploidy is assumed to be the quantitative equivalent of chromosomal aneuploidy [6]. The detection of DNA stemline aneuploidy corresponds to the detection of malignant cells [7,8], a technique that can be performed automatically and objectively [9].

Studies have shown that identifying DNA aneuploidy in squamous epithelium can lead to an earlier detection and diagnosis of OSCC by up to two years [8]. Since this non-invasive procedure is well tolerated by patients, OSCC could be detected at an earlier stage, which could significantly increase the chance of recovery and thereby reduce the burden on the healthcare system.

Although DNA aneuploidy is an accepted biomarker for malignancy [10], the effectiveness of the procedure is still controversially discussed [11,12]. Considering various studies, pooled sensitivities and specificities of 55–100%, differently applied methods of DNA measurement (image cytometry vs. flow cytometry), small cohort sizes <200 and inconsistently applied definitions of aneuploidy resulted in the fact that there is limited evidence [12,13]. Therefore, a uniform implementation of the procedure, clearly defined parameters and statistical power are necessary to obtain an objective statement on the accuracy of the procedure, as an additional tool.

PICO statement: This paper presents a prospective study examining outcomes from November 1997 to August 2010 at our oral medicine unit. Parts of the cytological examination and DNA ploidy measurement data used in this article were preliminary published in another study [14].

The outcome parameters were defined as follows: In view of the aforementioned limitations, the aim of this study, thus, was to statistically verify the diagnostic accuracy of DNA-ICM as an adjunctive tool to oral brush biopsy-based cytology samples to detect severe oral epithelial dysplasia, carcinoma in situ or OSCC in a large series of various clinical lesions in 602 cases. In addition, we tested, by logistic regression, whether DNA-ICM is an independent discriminator for the detection of malignancy.

## 2. Methods and Material

### 2.1. Patients and Cell Collecting Procedure

This study was approved by the ethical committee of the University Hospital Leipzig, Germany (no. 1272002), and all participating subjects were informed and signed consent. In total, 467 patients with clinical aspects of oral lesions visiting Leipzig University Hospital during 1997–2010 were enrolled in this study. Each lesion was first inspected visually, palpated and classified into a clinical diagnosis (Table 1). Staining of the specimens, cytological diagnoses and DNA-ICM were performed at the Institute of Cytopathology, Heinrich Heine University, Düsseldorf, Germany, as part of patient care.

For the examination, cell material from the suspected lesion was first removed in a rotating manner using brush biopsy (Cytobrush GT). The cells and cell clusters obtained were smeared and fixed with alcohol spray on a glass slide for conventional cytology. This obtainment procedure was repeated up to four times. Following the alcoholic fixation of the preparations, a special cytological routine staining using Papanicolaou (PAP) was carried out. Tissue was biopsied from the same location of the brushing, fixed with formalin, embedded in paraffin and subjected to histopathological examination.

In our cross-sectional study, approximately five swabs from each lesion were cytologically examined by experienced cytopathologists, and the specimens were classified as negative (1), doubtful (2), suspicious (3), and positive (4) for tumor cells. The DNA content of each sample suspected or indicative of malignancy (only group 2–4) was determined on the identical slide by DNA ploidy analysis with DNA-ICM and compared to either the histological results of scalpel biopsy or clinical follow-up.

The inclusion criteria were as follows:(a)Primary clinical diagnosis of OPMD or verified diagnosis of OSCC;(b)Both brushing and biopsy samples were obtained;(c)DNA content analysis was completed by DNA-ICM;(d)Histopathological examination by pathologist, or for the cytologically negative controls, clinical follow-up on average of 60 months by experienced oral surgeons.

All oral lesions of the period of investigation with not negative cytological diagnoses (categories 2–4) and available DNA ploidy results were included. In addition, 204 randomly selected samples with no evidence of malignancy in the clinical follow-up, which were cytologically assessed as negative, were included and examined as negative controls by DNA-ICM (Figure 1).

The results of cytology and DNA-ICM were compared to the histological and/or clinical follow-up. Severe oral epithelial dysplasia, cytological diagnosis, evidence of DNA aneuploidy with a consistent clinical course (i.e., OSCC therapy, definitive imaging, or palliative care), or histologically confirmed carcinoma in situ and OSCC was defined as positive follow-up. Histology of low-grade/intermediate oral epithelial dysplasia, benign histology, or a negative clinical follow-up of 2 years in the same oral region was defined as negative follow-up.

### 2.2. Technical Approach of DNA-ICM

Slides prestained with PAP were examined microscopically; cells of interest were marked with a felt pen on the coverslip and photocopied to ensure marked cell clusters could be found after uncovering during the Feulgen staining process. Feulgen staining was used for quantitative staining of nuclear DNA. The QUIC-DNA system (Tripath, Burlington, NC, USA) was used in combination with a conventional light microscope Axioplan 2 (Zeiss, Jena, Germany) for photometric analysis of the integrated optical density (IOD) of the cell nuclei.

The mean DNA content of >30 cytologically normal epithelial cells or lymphocytes was measured as an internal reference. Around 300 randomly selected, atypical cells from the cell population examined were measured. The reference absorbance was normalized to the 2c value of the reference cells. Coefficients of variation of reference cells were below 5%. The European Society for Analytical Cellular Pathology (ESACP) standards and guidelines for DNA-ICM were followed [6].

### 2.3. Criteria of Aneuploidy

Most human tumors are characterized by a numerical and/or structural chromosome aberration. The DNA content of 2c (c = content) of physiological epithelia is determined by measuring around 30 reference cells (normal squamous epithelial cells) using their average IOD value.

The DNA histogram of a normal proliferating cell population shows a first peak at 2c and a second at 4c (Figure 2).

If the following criteria were met, the result was interpreted as aneuploid.

According to the ESCAP [6], DNA aneuploidy is recognized either in a deviation of the DNA stemlines or in appearance of so-called “rare events”.

DNA stemline aneuploidy: The usual precision of recent DNA image cytometric measurements should at least allow DNA stemlines to be identified as abnormal (or aneuploid) if they deviate more than 10% from the diploid (2c) or tetraploid region (4c), i.e., if the modal values of DNA stemlines are outside 2c ± 0.2c or 4c ± 0.4c (examples: Figure 3 and Figure 4).

**Figure 2 cancers-14-05828-f002:**
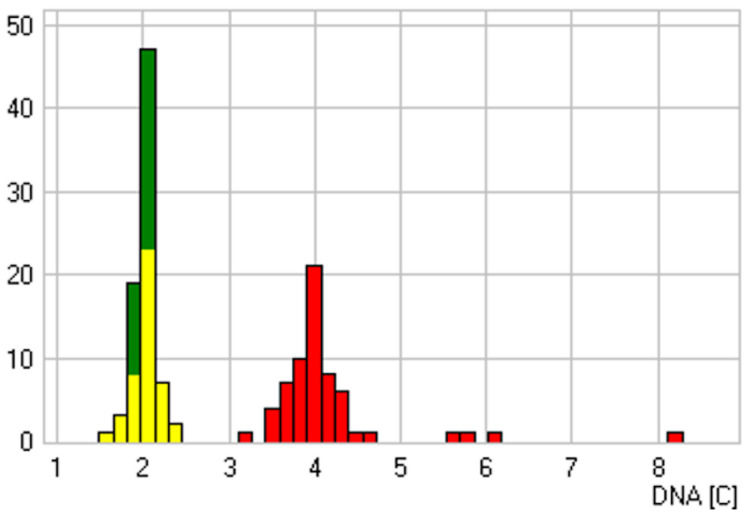
Sample with a doubtful cytological diagnosis and euploid–polyploid DNA histogram. Two euploid DNA stemlines at 2c and 4c and single cells with DNA contents up to 8.3c are shown. DNA content in c-units (*x*-axis), number of cells (*y*-axis), reference cells (green), analysis cells (red), biggest stemline (yellow). In this patient case, a 66-year-old female patient showed an ulceration on the right edge of the tongue, which was histologically diagnosed as granulation tissue and healed after 6 weeks without irritation.

**Figure 3 cancers-14-05828-f003:**
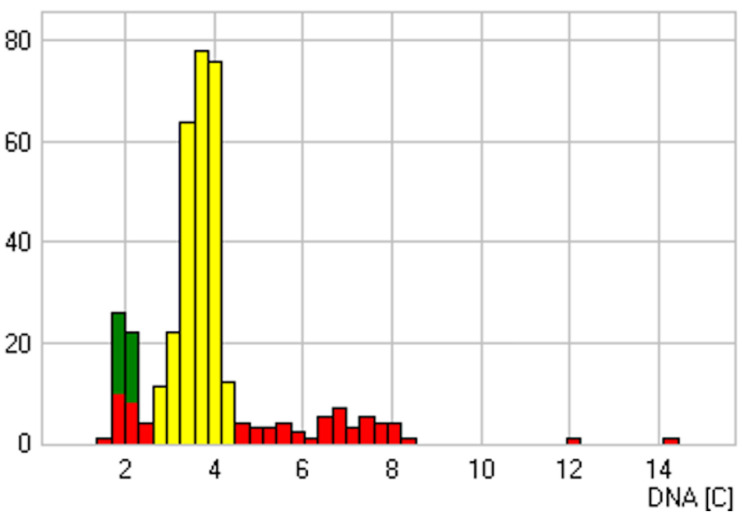
Sample with a suspicious cytological diagnosis and presence of DNA aneuploidy as a sign of malignant transformation. An atypical stemline at 3.52c is shown with an associated doubling peak and two cells with a DNA content >9c. DNA content in c-units (*x*-axis), number of cells (*y*-axis), reference cells (green), analysis cells (red), biggest stemline (yellow). In the follow-up of this sample, a second carcinoma of the floor of the mouth was diagnosed in an 86-year-old male patient.

**Figure 4 cancers-14-05828-f004:**
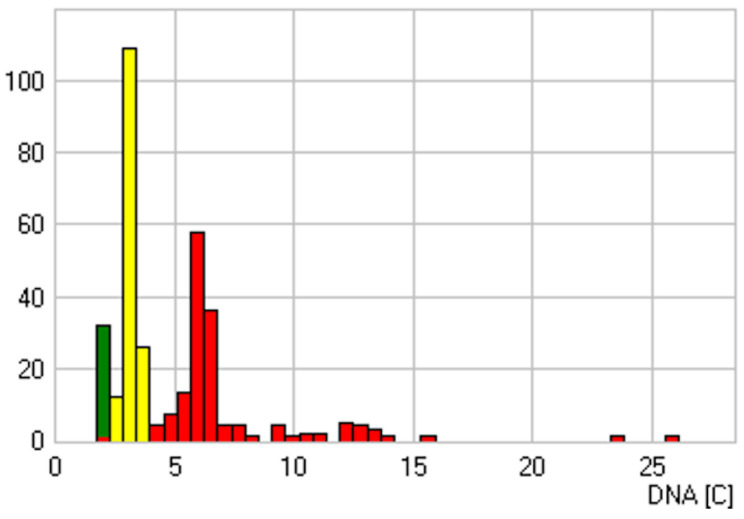
Sample with a positive cytological diagnosis and DNA aneuploidy as a sign of malignant transformation. Atypical stemline distribution at 3.24c and 6.5c. In addition, several cells with a DNA content >9c are shown. DNA content in c-units (*x*-axis), number of cells (*y*-axis), reference cells (green), analysis cells (red), biggest stemline (yellow). This sample is a carcinoma of the soft palate in a 68-year-old male patient.

2.DNA single cell aneuploidy: Rare events in DNA histograms are abnormal cells often called 5c or 9c exceeding events, having a nuclear DNA content higher than the duplicate or quadruplicate region of a normal G1/G0 phase population, i.e., not belonging to the G2M phase. Accordingly, we defined “single cell aneuploidy” as the occurrence of at least one cell with a DNA content of >9c (example: Figure 3 and Figure 4).

### 2.4. Statistical Analysis

Grouping of cytopathology diagnoses was performed for the application of logistic regression. A negative diagnosis was declared as 1 = negative, and the above-mentioned cytopathological diagnostic categories 2–4 have been combined as 2 = positive. For DNA-ICM, a case was scored as “negative” if no DNA aneuploidy could be detected and as “positive” for evidence of DNA aneuploidy. Then, 95% confidence intervals were given for the sensitivity, specificity and AUC measurements. All analyses were performed with SPSS (Version 26; IBM Corporation, Armonk, NY, USA) and STATA (Version 16., StataCorp LLC, College Station, TX, USA) for Windows.

## 3. Results

### 3.1. Enrolled Patients

A total of 602 samples of 467 patients were included in this study: 375 samples of the study population were women (62.3%), and 227 samples were men (37.7%) (Table 1). At the time of sampling, the mean age for women was 57.73 years and for men 63.65 years.

Upon final histopathological examination, 308 patients were diagnosed with OSCC and 90 patients with varying degrees of epithelial dysplasia. Independently, an additional 204 samples with no indication of malignancy were examined, which also proved to be negative in the clinical follow-up. OSCC and epithelial dysplasia occurred mainly on tongue (30.3%), floor of mouth (21.54%), and buccal mucosa (17.7%). The mean age of the total cohort for OSCC was 59.05 years (SD 12.17) and 60.93 years (SD 13.63) for epithelial dysplasia. These values are two-tailed (0.075) by *t* test with Satterthwaite correction for heterogeneous data. From 308 samples diagnosed as OSCC (Table 2), 76.20% were seen in women, mean age 57.82, and 23.8% were seen in males, mean age 63.07. The age correlation between the male and female cohort was significant (Chi² N-1-Test).

### 3.2. Cytological Examination

In a cytological examination, the specimen is examined by a pathologist under the light microscope for abnormalities in cell structure, especially of the nucleus. A diagnosis is made of negative, doubtful, suspicious and positive for tumor cells. For further statistical calculations, a dichotomization into positive and negative was made (see Statistical Analysis). In comparison to the histological biopsy/clinical follow-up of the patients with lesions suspicious or indicative for malignancy, 308 cases were identified as true positive, 204 cases as true negative, 90 cases as false positive and no cases as false negative. This results in a sensitivity of 100% (95% CI 98.8–100.0%), specificity of 69.4% (95% CI 63.8–74.6%), PPV = 77.4% (95% CI 73.0–81.4%) and NPV = 100% (95% CI 98.2–100%) (Table 3). The area under the curve was 0.85 (95% CI 0.82–0.87) (Table 3).

### 3.3. DNA Image Cytometry

In the field of diagnostic studies, the AUC serves as an overall measure of a diagnostic test’s accuracy (Figure 5). With the criteria for DNA aneuploidy mentioned above, the >10% deviation of DNA stemline from physiological values and/or cells >9c, the sensitivity for detecting severe epithelial dysplasia or carcinoma within OPMD due to DNA ICM was 93.5% (95% CI 90.1–96.0%), and specificity was 98.0% (95% CI 95.6–99.2%). The recorded positive predictive value was 98.0%, and the negative predictive value was 93.5% (Table 3). A total of 20 cases with positive follow-up were missed by DNA-ICM, and 6 cases of OPMD were false positives. Using post hoc power analysis, it can be evaluated that the cohort size of 602 cases is statistically significant at the 5% level (alpha level = 0.05). A comparison of cytology and DNA ploidy analysis of the 602 specimens is shown in Table 4. The frequencies of aspects of DNA ploidy in relation to the biological behavior of the lesion are given in Table 5.

### 3.4. Logistic Regression

Logistic regression was applied to evaluate odds ratio (OR) and the association between binary target variables. First, we analyzed the association among cytological examination and histological biopsy using bivariate analysis. Odds ratio for the cytological examination as a sole predictor was 171.11 (95% CI 61.87–473.20) (Figure 6). Using subsequent multiple logistic regression, we included DNA image cytometry and obtained odds ratio (cytology) = 11.90 (95% CI 3.94–35.88) and OR (DNA-ICM) = 201.6 (95% CI 78.42–518.22, *p* = 0.001). Cytological examination predicted AUC of 0.85 (95%CI 0.82–0.87) and AUC (DNA-ICM) = 0.96 (95%CI 0.94–0.97), respectively (Figure 5). The areas under the curve are significantly different (Chi^2^(1) = 112; *p* < 0.001).

## 4. Discussion

In our study, the value of DNA ploidy analysis using DNA-ICM as an additional diagnostic screening tool for the detection of severe oral epithelial dysplasia and OSCC in clinically suspect oral lesions including OPMD was examined and rated with a high accuracy. Our prospective study was carefully designed to eliminate other possible variables that have led to controversial literature statements in the past, such as lack of heterogeneity of included clinical lesions, small study population, and inconsistent definition of aneuploidy by following the guidelines according to the ESACP consensus reports [6].

Aneuploidy is a meaningful, recognized biomarker that indicates numerical chromosome changes and thus a developmental anomaly, which in addition is often used in cancer diagnostics [15,16,17]. The cytometric equivalent of chromosomal aneuploidy, determined by DNA-ICM, is DNA aneuploidy [6].

Although the oral cavity is a highly visible area of the human body, too little visual attention is paid to it in practice. Taking brush biopsies offers the dentist the opportunity to have questionable changes in the oral mucosa examined in an uncomplicated way. This procedure takes only a few minutes during treatment, and the cost of materials is vanishingly small compared to the cost of therapy for OSCC in advanced stages. By examining by means of brush biopsies, the treating dentist can help to minimize the secondary time loss of tumor patients until adequate therapy is available and can significantly improve the prognosis [18]. Furthermore, brush biopsies make it possible to examine larger altered areas, whereas with conventional biopsies, the treating dentist has to decide on a limited area. By using brush biopsies, the referral hurdle to an oral surgeon is eliminated and possible bleeding, wound healing disorders or the consideration of anticoagulants in the patient can be neglected with brush biopsy.

There are several already communicated advantages of brush biopsy-based oral cytology. Gupta et al. has found that it is a way of early detection, and as a result, the time between diagnosis and treatment can be significantly reduced [18]. Furthermore, it is a painless, non-invasive technique that is well accepted by patients.

There are various indications for the additional use of brush biopsies. The noninvasive, painless use of brush biopsies results in high patient compliance in contrast to conventional biopsy, which is why they are useful for monitoring oral lesions in general [19], for patients with allergies to local anesthesia and thus limited compliance for biopsy [20], and for patients who require frequent follow-up, such as in Fanconi anemia [21].

In various studies investigating the use of DNA ploidy analysis or DNA-ICM of the last decades, there is a wide range for sensitivity (16–100%) and specificity (66–100%). One of the main reasons is the inconsistent definition of DNA aneuploidy across studies. In the study by Pektas et. al, which indicates a sensitivity of 16%, DNA aneuploidy was defined if the peaks in the histogram were at 3c, 5c, 7c and 9c or if the number of nuclei with a DNA content of more than 5c or 9c was over 1% [22]. With this definition, only 2 out of 12 OSCC could be detected, resulting in low sensitivity. In contrast, the study by Maraki et al. reached a sensitivity of 100%. The definition used corresponds to the definition from our study design [23].

In terms of specificity of our study, DNA-ICM was 92.3% (Table 3), which was much higher than the specificity of conventional cytology, which was 69%. Early detection of dysplastic squamous cells is highly desirable, and accordingly, the doubtful and suspicious cytology in this study was assigned to the statistical category of “positive,” which allowed for maximum sensitivity but at the expense of lower specificity. With this assignment of doubtful and suspicious cytology, the sensitivity of conventional cytology is 0.8% higher than that of DNA-ICM. However, this is only a statistical effect, as the determination of DNA ploidy provides a more definitive result for a clinical measure than does doubtful or suspicious cytology. In general, the probability of malignancy is considered to be approximately 30% and 70% for equivocal and suspicious cytology, respectively [2,3,4,24].

In our study, the positive predictive value of DNA-ICM was 98.0% and the negative predictive value was 93.5%. A total of 20 cases of severe epithelial dysplasia or OSCC were assessed as false negative. One of the reasons could be the number of cells examined. On average, at least 300 cells should be measured [25]. In 16 of the 20 cases above, only 57–186 cells were examined instead of the required minimum of 300 cells. Furthermore, a possible misjudgment can be attributed to the fact that cell overlaps can occur, which make it difficult to measure their DNA content [19].

Inflammatory exudate and necrotic deposits on the lesion can lead to an insufficient number of evaluable cells to be obtained for examination, thus increasing the false-negative rate. Therefore, in this clinical condition, cell harvesting should be avoided, and a brush biopsy should be performed only after this condition has passed [19].

Finally, using logistic regression, a clear effect for the sole predictor of cytological examination as an examination method could be demonstrated. However, with the addition of ploidy examination as a second predictor, the effect of cytology is reduced, whereas the effect of DNA-ICM is very clearly recognizable.

Especially for cancer screening, the sample size is essential in order to be able to achieve statistical power. To the best of our knowledge, the sample size (*n* = 602) was the largest-scale series evaluating DNA-ICM following the definitions of DNA aneuploidy consented by the ESACP [6].

We are aware of the limitations of the study design and recognize that handling brushing devices as well as interpreting DNA-ICM results require a certain amount of skill. The proportional distribution of the examined clinical lesions that are enriched for (pre)malignant ones is inconsistent with the frequency in daily practice. This is due to the fact that manual DNA-ICM is mostly performed subsequent non-negative cytology. A majority of the lesions examined were OSCC, leukoplakia and oral lichen, resulting in statistically underpowered results for minor subtypes of OPMD. For statistical calculation of the diagnostic accuracy of the DNA-ICM, the DNA ploidy of 204 randomly selected samples of the cytological classification “negative” (1) was examined. Their clinical follow-up of an average of 60 months proved to be “definitely negative”. In conclusion, it can be confirmed that DNA aneuploidy is a marker for malignancy in epithelial oral cells.

Nevertheless, our study design is a cross-sectional study. A longitudinal study would have to be carried out for a valid examination regarding the use of brush biopsies as a follow-up screening tool and therefore the impact on disease progression.

Recently, liquid biopsy, circulating biomarkers, and noninvasive sampling have attracted great interest from the scientific community. Regarding the use of brush biopsies as a noninvasive diagnostic method, one study has shown that liquid biopsies provide a faster result with the same sensitivity and specificity because only one sampling is required [26]. Another research focus is the determination of biomarkers as a diagnostic tool for the detection of malignant changes in the oral cavity. To date, however, no sole, unique biomarker has been identified, but the use of brush biopsies to quantify cancer-specific methylation characteristics has been shown to be helpful in establishing risk stratification schemes for OSCC [27,28,29]. In another study, brush biopsies were used to screen patients’ saliva for the presence of *p*-53 codon 63 mutations, which could serve as predictors of OSCC [30]. In conclusion, by means of our study, DNA image cytometry was shown to be a highly sensitive, noninvasive, and patient-friendly method in addition to conventional cytology for detecting potentially malignant oral lesions with malignant transformation requiring clinical intervention. Widespread use as a routine screening method could improve early detection of suspicious lesions, increase OSCC survival, and thus reduce the burden on the healthcare system.

## 5. Conclusions

In summary, this study has shown that DNA ploidy analysis based on conventional specimens of oral brush biopsies is a highly sensitive, non-invasive, patient-friendly method that should be considered as an additional diagnostic tool for detecting malignant changes in the oral cavity.

## Figures and Tables

**Figure 1 cancers-14-05828-f001:**
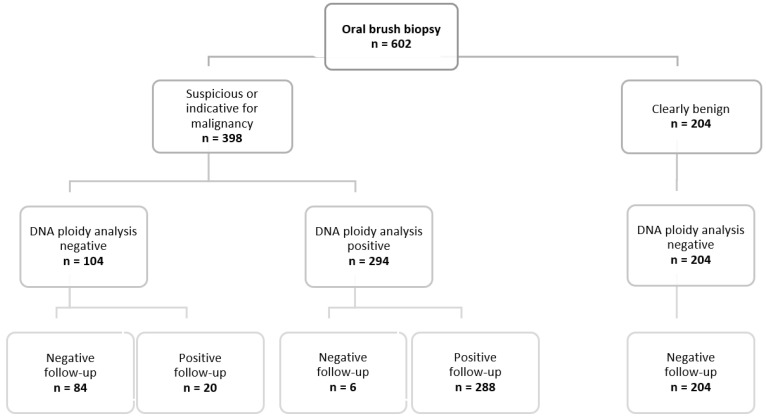
Presentation of the results of the oral brush biopsy, the subsequent DNA ploidy analysis and their follow-up outcome.

**Figure 5 cancers-14-05828-f005:**
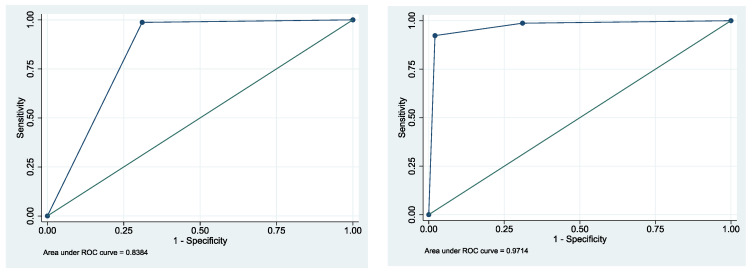
ROC (receiver operating characteristics) curves for cytological examination (**left**) of the cells and DNA image cytometry (**right**).

**Figure 6 cancers-14-05828-f006:**
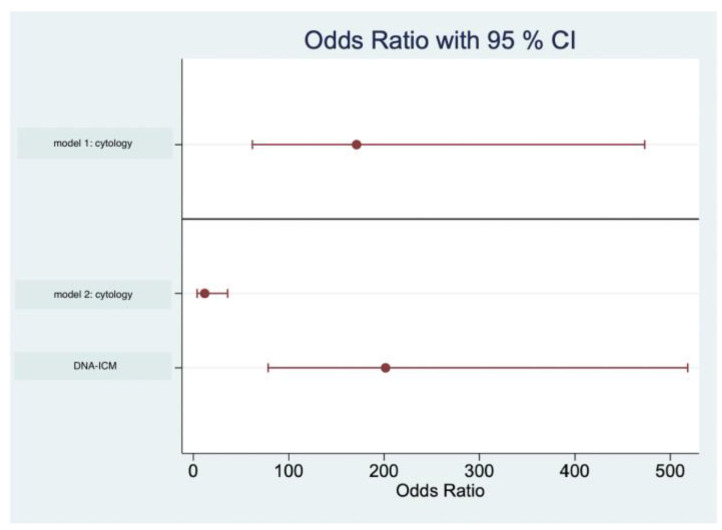
Model 1: calculation of the logistic regression with the cytological examination as the sole predictor (OR = 171.11). Model 2: addition of the second predictor. DNA-ICM: increased association between DNA-ICM and detection of malignant cells (OR = 201.6) in addition to decreased association for cytological examination (OR = 11.90).

**Table 1 cancers-14-05828-t001:** Distribution of the most frequently occurring clinical diagnoses according to the gender of the subjects.

Clinical Diagnoses		Gender		Total
		Female	Male	
Oral Leukoplakia	number of cases	68	44	112
	%	60.7	39.3	
Oral Lichen planus	number of cases	8	27	35
	%	22.9	77.1	
Lichen erosivus	number of cases	5	19	24
	%	20.8	79.2	
OSCC *	number of cases	228	73	301
	%	75.7	24.3	
Ulceration	number of cases	31	18	49
	%	63.3	36.7	
Other lesions **	number of cases	40	41	81
	%	49.4	50.6	

* oral squamous cell carcinoma. ** for example: pemphigus vulgaris, canker sores, candidiasis, smokers keratosis, herpes simplex, etc.

**Table 2 cancers-14-05828-t002:** Overview of the classification of the investigated OSCC in their TNM classification, regarding tumor extent, lymph node involvement and presence of distant metastases.

T-Status	(*n*)	N-Status	(*n*)	M-Status	(*n*)
Tx	55	Nx	55	Mx	55
Tis *	1	N0	139	M0	186
T1	84	N1	43	M1	71
T2	85	N2a	14		
T3	37	N2b	42		
T4a	46	N2c	12		
T4b	0	N3	3		

* Including severe dysplasia and carcinoma in situ.

**Table 3 cancers-14-05828-t003:** Sensitivity, specificity, positive and negative predictive values (PPV and NPV) of cytology and DNA aneuploidy for oral potentially malignant disorders (OPMD) and OSCC.

Method	Specificity	Sensitivity	PPV	NPV	AUC	95% CI (AUC)
Conventional Cytology	69.4%	100.0%	77.4%	100%	0.85	0.82–0.87
DNA-ICM	98.0%	93.5%	98.0%	93.5%	0.96	0.94–0.97

**Table 4 cancers-14-05828-t004:** Overview of the results of the cytological examination and the DNA-ICM in comparison.

Cytological Examination	DNA Ploidy Analysis with DNA-ICM	
	DNA Aneuploidy	No DNA Aneuploidy
positive	222	6
suspicious	49	14
doubtful	23	84
negative	0	204

**Table 5 cancers-14-05828-t005:** Frequency of met criteria for DNA ploidy in relation to the biological behavior of lesions.

Aspects of DNA-Ploidy	Biological Behavior of the Lesion	
	Malignant	Benign
Normal stemline [1.8 < c < 2.2] ^x^ x = 1,2,3	20	288
Atypical stemline [1.8 > c > 2.2] ^x^ x = 1,2,3	227	5
1	78	3
2	118	2
>2	31	
Cells > 9c	301	4
1 to 3	112	3
4 to 10	86	1
>10	103	0
Atypical stemline and cells > 9c	222	3

## Data Availability

Data presented in this study are available upon request from the corresponding author. Data are not publicly available for privacy reasons.

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
