# Peer review of "Evaluation of Static DNA Ploidy Analysis Using Conventional Brush Biopsy-Based Cytology Samples as an Adjuvant Diagnostic Tool for the Detection of a Malignant Transformation in Potentially Oral Malignant Diseases: A Prospective Study"

_cancers, 2022, doi:10.3390/cancers14235828_

Round 1

Reviewer 1 Report

Congratulations on the article! It is very important that we have other diagnostic options,

especially non-invasive techniques, and that they maintain (or even increase) the fidelity of what

has already been found in cytology. This is very important, mainly in OPMD. However, here are

some simple suggestions and questions.

- In the introduction, a suggestion is to change when you quote "Oral potentially malignant

disorders (OPMD), including leukoplakia, erythroplakia, oral lichen planus etc. are mucosal

diseases that have a risk of malignancy being present at the time of initial diagnosis or a future

date." To: "Oral potentially malignant disorders (OPMD), including mainly leukoplakia,

erythroplakia and oral lichen planus are mucosal diseases that have a risk of malignancy being

present at the time of initial diagnosis or a future date."

- Figure 1: Uppercase in Sample...

Figure 2: Uppercase in Sample...

Figure 3: Uppercase in Sample...

- In table 2, what is the possible reason for the large difference between the specificity of

conventional cytology and DNA-ICM? (If possible talk more about in the discussion).

- In flow chart 1, please uppercase all the beginnings of terms, to follow the pattern you put in

"Oral brush biopsy".

- When in the results you talk about the odds ratio, keep a standard in the spacing of values ​​and

symbols (=), such as: odds ratio (cytology) = 11.90 (95%CI 3.94-35.88). These are details, but it

makes it easier to visualize the results.

- In figure 5: Why such a big difference between the OR of the specific cytology, when alone,

models between model 1 and 2?

- In the discussion, I think it is necessary to address more the advantages of using the technique

studied in comparison with the traditional ones, in addition to the obvious advantages.

- Add more references on the subject.

Reviewer 2 Report

Bechstedt and colleagues presented a research article aimed at evaluating the diagnostic potential of the analysis of DNA ploidy starting from cytological slides obtained through non-invasive oral brushing. Overall, the research idea is very interesting, however, the experimental design is very simple and some relevant information was not considered. Please address the following comments:

1) Did the authors evaluated also the loss of DNA (also leading to aneuploidy)? Please, clarify;

The authors presented only general data about patients’ lesions. For example, they indicated the percentages of patients affected by OSCC, lichen planus or dysplasias. Please be more detailed with the description of the clinical pathological features of patients (at least for OSCC patients). You should add an additional table reporting the tumor staging, the presence of metastasis, positive lymph nodes, and all the relevant information;

The results obtained are interesting, however, the authors have to add more information to their data. For example, are there any correlations between the severity of DNA ploidy and the use of tobacco, alcohol abuse or other well-known risk factors for oral cancer? These data are fundamental. In particular, it would be interesting to evaluate if benign lesions with low or moderate DNA ploidy are due to such risk factors;

In the Discussion section, the authors should better describe the importance of non-invasive diagnosis in oral cancer. More recently, the use of liquid biopsy, circulating biomarkers and non-invasive sampling has attracted great interest in the scientific community. Please add a brief description of the potentiality of oral brushing, liquid biopsy and non-invasive diagnostic strategies for both neoplastic and benign oral lesions. For this purpose, please see:

- PMID: 36005828

- PMID: 31752196

- PMID: 34930473

- PMID: 35153426

Round 2

Reviewer 1 Report

Accepted

Author Response

Thank you for your comments and acceptance of the paper. 

Reviewer 2 Report

The authors well-addressed almost all of my previous comments. I am not sure that the Refs from 4 to 8 are appropriate for the description provided in the Discussion section (from line 358 to line 373). Please improve this part and see the suggestions in my previous comment No. 4.

Author Response

Thank you for the feedback. We had technical problems in the consecutive numbering of the source directory. All sources have been put into the correct order again, so this error should be fixed. We will upload the revised manuscript again.